# Biomimetic Superhydrophobic Materials through 3D Printing: Progress and Challenges

**DOI:** 10.3390/mi14061216

**Published:** 2023-06-08

**Authors:** Haishuo Liu, Zipeng Zhang, Chenyu Wu, Kang Su, Xiaonan Kan

**Affiliations:** 1School of Mechanical Engineering, Shijiazhuang Tiedao University, Shijiazhuang 050043, China; lhs3707@163.com; 2College of Polymer Science and Engineering, Qingdao University of Science and Technology, Qingdao 266042, China; zzp13165436954@163.com; 3Qingdao Institute for Theoretical and Computational Sciences, Institute of Frontier and Interdisciplinary Science, Shandong University, Qingdao 266237, China; w@sdu.edu.cn

**Keywords:** biomimetic, superhydrophobic, 3D printing, fabrications, applications

## Abstract

Superhydrophobicity, a unique natural phenomenon observed in organisms such as lotus leaves and desert beetles, has inspired extensive research on biomimetic materials. Two main superhydrophobic effects have been identified: the “lotus leaf effect” and the “rose petal effect”, both showing water contact angles larger than 150°, but with differing contact angle hysteresis values. In recent years, numerous strategies have been developed to fabricate superhydrophobic materials, among which 3D printing has garnered significant attention due to its rapid, low-cost, and precise construction of complex materials in a facile way. In this minireview, we provide a comprehensive overview of biomimetic superhydrophobic materials fabricated through 3D printing, focusing on wetting regimes, fabrication techniques, including printing of diverse micro/nanostructures, post-modification, and bulk material printing, and applications ranging from liquid manipulation and oil/water separation to drag reduction. Additionally, we discuss the challenges and future research directions in this burgeoning field.

## 1. Introduction

Being a unique phenomenon, superhydrophobicity has assisted organisms to survive under harsh natural conditions for billions of years [1,2,3,4,5]. Superhydrophobicity refers to objects with high water contact angles (CAs) (>150°). For instance, a lotus leaf can repel water and keep clean in sludge, cactus spines and desert beetles can collect water in dry environments, water striders can walk freely on water surfaces, and the wings of many butterflies also have anisotropic superhydrophobic properties [6,7,8,9]. In nature, two main superhydrophobic effect can be found. One is the “lotus leaf effect”, referring to surfaces with water CAs larger than 150° and contact angle hysteresis (CAH) of less than 10° [10]. These surfaces show superhydrophobic property with low adhesion, which can be attributed to a combination of hierarchical micro/nanostructures and wax with low surface energy. The other is the “rose petal effect”, referring to surfaces with water CAs larger than 150° and contact angle hysteresis (CAH) of more than 10° [11]. These kinds of surfaces show superhydrophobic property with high adhesion and, although possessing unique fractal micro/nanostructures, no low surface energy coatings can be found, leading to high adhesion of water droplets.

Learning from nature, extensive studies have been carried out on fabricating biometric superhydrophobic materials during recent decades, covering from theoretical wetting models and different fabrication strategies to diverse kinds of applications [12,13,14]. To further reveal the wetting regimes, several famous models were proposed, including Wenzel models for complete wetting [15], Cassie–Baxter models considering surface roughness and heterogeneity [16], and intermediate models between these two states [17,18]. In addition, to mimic natural analogues, numerous kinds of strategies have been developed for fabrication of superhydrophobic materials, such as chemical etching, spray coating, electrochemical deposition, lithography pattering, sol–gel processing, etc. [19,20,21]. The main construction guideline is a combination of surface roughness with low-energy materials. The fabricated multifunctional materials have shown great potential in various applications fields, such as anti-icing, water/oil separation, directional liquid transportation, drag reduction, etc. [22].

Among all these fabrication techniques, 3D printing, also known as additive manufacturing (AM) or free-form fabrication, has attracted extensive attention due to its special role in construction of artificial superhydrophobic materials [23,24]. The main advantages of 3D printing over others are the fast construction of complex materials without the use of a template with low cost and high precision. During the recent years, several kinds of 3D printing techniques, such as stereolithography (SLA), digital light processing (DLP), fused deposition modeling (FDM), and direct ink writing (DIW), have been reported for the successful preparation of superhydrophobic materials [25,26,27,28]. Based on these strategies, different biomimetic micro/nanostructures such as re-entrant structures [29,30,31] and eggbeater analogues [32,33] can be printed, and the printed objects can be further modified to improve water repellency [34,35,36]. More recently, bulk superhydrophobic objects have been fabricated directly through 3D printing [37,38]. Although great progress has been made in these areas, urgent challenges still remain. One of the biggest is the lack of structure–function relationships as clear guidelines. In consequence, challenges and opportunities coexist in 3D printed superhydrophobic materials, calling for further advancements in this field.

In this minireview, we will summarize the recent progress in biomimetic superhydrophobic materials fabricated through 3D printing, mainly focusing on wetting regimes, different fabrication techniques, and various applications of the printed objects (Figure 1). Finally, conclusions and outlooks are presented for a comprehensive discussion of the challenges and future research directions in this field.

## 2. Natural Superhydrophobic Surfaces and Wetting Regimes

In nature, many kinds of plants and animals have superhydrophobic surfaces, attracting extensive attention due to their unique properties, such as self-cleaning, water-proofing, uniaxial water transport, etc. Although the apparent CAs of these surfaces are similar, the CAHs may be different considering the chemical compositions of these surfaces. Additionally, the important “lotus leaf effect” and “rose petal effect” are proposed according to the water repellency/adhesion. The exploring of wetting regimes can provide important guidance for the fabrication of artificial superhydrophobic materials and much effort has been made in this field. Moreover, to further reveal the wettability mechanism, several models have been developed, originating from Young’s equation, including the Wenzel model, Cassie–Baxter model, and some new models. Still, debates exist on whether these models have explained wettability explicitly.

### 2.1. Natural Superhydrophobic Surfaces—“Lotus Leaf Effect” and “Rose Petal Effect”

As is well known, a lotus leaf can withstand muddy environments due to its self-cleaning property. The apparent CAs of a lotus leaf can be more than 150° and CAHs can be less than 5° [39]. As a result, water can roll down the leaf easily. During the rolling process, the dirt on the leaf can be absorbed into the water drop, so-called liquid marbles are thus formed, and the surface energy can be minimized after adsorption. Numerous efforts have been made in revealing this mechanism [40]. At first, this phenomenon was ascribed to the wax coating and microstructures by Barthlott and Neinhuis [41]. In 2002, Feng et al. revealed that the surface of a lotus leaf is composed of large amounts of micropapillae and nanostructures on top of them [10]. These hierarchical micro/nanostructures play a vital role in superhydrophobic and self-cleaning functions (Figure 1a,b). In addition to the lotus leaf, some other plant leaves were reported to have similar properties, such as *Colocasi esculenta* leaves [42], *Salvinia* leaves [43], and so on. Furthermore, these water-repellent structures can motivate design of surfaces which can be applied in anti-icing and anti-fogging [44].

Similar to the lotus leaf, the rose petal also exhibits superhydrophobic property. However, different from the former, water droplets can adhere on top of the petal firmly [45,46]. The CAH is large in these structures where the pinned water will not roll away until complete evaporation. This can help animals or plants to survive in dry environments. As can be seen from closer observation through SEM, periodic microarrays with an average diameter of about 20 μm and height of 10 μm can be found on the surface, covered with nanosized wrinkles [47] (Figure 2a,b). Jiang et al. revealed that the superhydrophobicity can be ascribed to the microstructures while the existence of the wrinkled nanostructures makes the contact area larger, leading to high adhesion of water [11]. Apart from rose petals, sunflower petals and Chinese Kaffir lilies also exhibit similar properties, showing superhydrophobicity with high adhesion [48]. In addition, the “rose petal effect” could prompt development of biometric superhydrophobic surfaces with high adhesion to retain liquids.

**Figure 1 micromachines-14-01216-f001:**
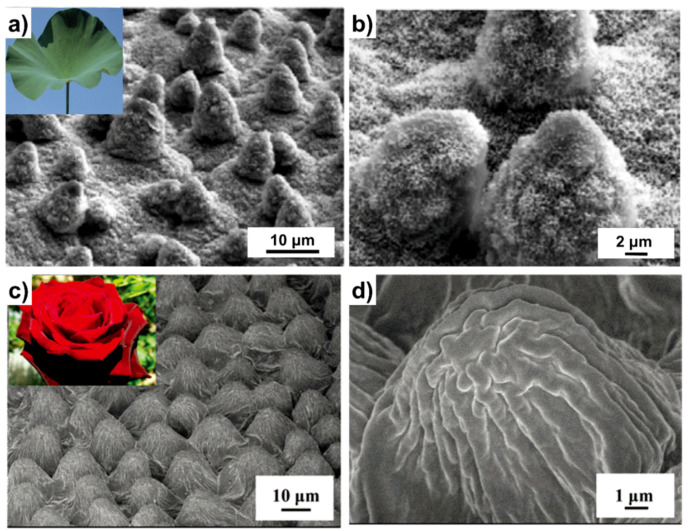
Natural superhydrophobic surface of (**a**) lotus leaf and (**b**) the SEM observation of micro/nanostructures. Reproduced from [40], copyright 2009, Royal Society of Chemistry. Natural superhydrophobic surface of (**c**) rose petal and (**d**) the SEM observation of micro/nanostructures. Reproduced from [47], copyright 2008, American Chemical Society.

**Figure 2 micromachines-14-01216-f002:**
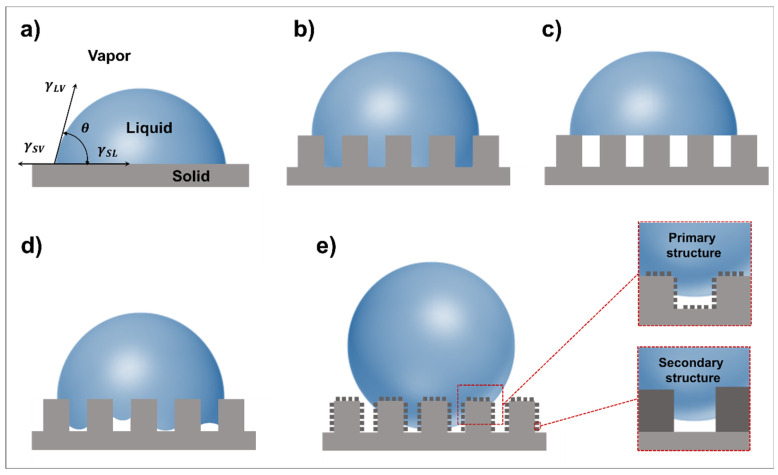
Wetting regimes of different models. (**a**) Description of Young’s equation. (**b**) Wenzel model. (**c**) Cassie–Baxter model. Intermediate model of micro/nanostructures with (**d**) one scale and (**e**) two scales.

### 2.2. Wetting Regimes

The surface wettability is mainly measured by CAs, and to further reveal the mechanism behind it, different models and equations have been proposed. The earliest and most famous equation is Young’s equation (Figure 2a), where the CA (θ) on a smooth surface is defined depending on the solid–vapor (SV), solid–liquid (SL), and liquid–vapor (LV) surface tensions (γ). When reaching equilibrium, the equation can be expressed as [49]:(1)$$\gamma_{SV}=\gamma_{SL}+\gamma_{LV}\cos\theta$$

As can be seen from the above equation, the CA is determined by interfacial energies, which are derived from thermodynamic equilibrium of the three phases’ (solid–liquid–vapor) interface free energy. However, Young’s equation is appropriate for ideal smooth surfaces and not suitable to describe realistic surfaces with roughness. Therefore, another two famous models, the Wenzel model and Cassie–Baxter model, were proposed accordingly.

#### 2.2.1. Wenzel Model

Compared with Young’s equation, the Wenzel model describes a complete wetting state considering surface roughness and it was proposed in 1936 [15]. In this model, a factor describing the surface roughness is introduced. The model assumes that water can completely contact a chemically homogenous surface (Figure 2b). The equation for the Wenzel model is expressed as follows:(2)$$\cos\theta_{w}=r\cos\theta$$
where $\theta_{w}$ and θ represent the Wenzel contact angle and Young’s contact angle, respectively, r is the surface roughness factor, equaling the fraction of true surface area to apparent surface area. The value of r is 1 for an ideal planar surface and more than 1 for a surface in reality. In consequence, as can be seen from the Wenzel equation, the introduction of roughness will increase both surface hydrophilicity and hydrophobicity to some extent. However, in the Wenzel model, it is supposed that the surface is homogenous and flat and the wetting is complete, which is not adequate for real surfaces. As a result, another model, the Cassie–Baxter model, was proposed to solve the above problems.

#### 2.2.2. Cassie–Baxter Model

In 1944, the Cassie–Baxter model was proposed to further investigate wetting regimes of heterogenous surfaces with roughness [16]. In fact, when water contacts a surface with certain roughness, it cannot fully enter the grooves due to existence of air pockets trapped in the grooves (Figure 2c). Therefore, the contact state is composite on such a surface. The equation for the Cassie–Baxter model is described as follows:(3)$$\cos\theta_{CB}=f_{1}\cos\theta_{1}+f_{2}\cos\theta_{2}$$
(4)$$f_{1}+f_{2}=1$$
where $\theta_{CB}$ stands for the Cassie–Baxter contact angle, $f_{1}$ and $f_{2}$ represent the apparent area ratio of surface 1 and surface 2. Commonly, surface 1 refers to the liquid/solid interface while surface 2 refers to the air pocket interface. As the air can only be trapped in the grooves along a solid surface, $\theta_{2}$ is 180° due to the non-wetting status. Therefore, Equation (3) can be simplified as follows:(5)$$\cos\theta_{CB}=f\left(\cos\theta +1\right)-1$$
where $\theta_{CB}$ still means the apparent contact angle, f refers to the fractional area of the liquid/solid interface. According to the equation, air pockets can be found in the “lotus leaf effect”, where the surface exhibits superhydrophobic property and low adhesion, verifying the importance of the Cassie–Baxter model in explaining actual wetting situations.

#### 2.2.3. Intermediate Models

Although the proposal of the above two models can fill the gap in wetting regimes to a great extent, they still cannot explain all wetting situations, especially those with a nanosized porous structure or hierarchical nano/microsized structure. As a consequence, several kinds of intermediate models have been developed during recent decades [17,18,50,51,52,53,54]. For instance, Miwa et al. discussed a mixed-wetting state at first. Luo et al. discovered the existence of a stable intermediate wetting state on patterned surfaces with microstructures [51]. Liu et al. systematically studied a combined Cassie–Baxter/Wenzel state on lotus leaves [52].

Here, an intermediate wetting state on surfaces with nano- or microstructures proposed by Nagayama et al. is specified as an example [18]. The results originated from molecular dynamic simulations. For a concavoconvex surface with micro- or nanostructures (Figure 2d), the CAs can be derived as:(6)$$\cos\theta_{CB}=f_{1}\cos\theta -f_{2}$$
where $\theta_{CB}$ is the constant contact angle, $f_{1}$ and $f_{2}$ are the fraction of the solid/liquid and liquid/vapor interface to the flat surface, respectively. If $f_{1}$ is 0, Equation (6) turns into the Wenzel model; if $f_{1}$ and $f_{2}$ fulfill Equation (4), the sum of the two equals 1, then Equation (6) is the Cassie–Baxter model. In other words, for partial wetting of an intermediate state, $f<f_{1}<r$, while $0<f_{2}<1-f$.

Meanwhile, apart from the above status, the contact angle of a surface with hierarchical nano/microstructures is also discussed in this report (Figure 2e), and the CAs can be described as:(7)$$\cos\theta_{CB}=f_{1}^{2}\cos\theta -f_{2}\left(f_{1}+1\right)$$

In this hierarchical structure, the wetting state of the primary structures is similar to that of the secondary structures, $f_{1}^{2}$ represents the area ratio of the solid/liquid interface to the apparent contact interface, $f_{2}f_{1}+f_{2}$ represents the area ratio of the liquid/vapor interface to the apparent contact interface.

#### 2.2.4. Debates

Nevertheless, despite all this progress made in wetting regimes, debates still exist in the exploration of ideal models to fully explain wettability in reality. For instance, Gao and McCarthy proposed that the wetting behavior of liquid on a structured surface is determined solely by interactions along the three-phase contact line (TCL) instead of the interfacial area within the contact perimeter [55]. The proposal was later proved by Liu et al. [52]. Erbil et al. tested the application range of the Wenzel model and Cassie–Baxter model by a simple method [56]. The results showed that the Wenzel equation was wrong for superhydrophobic samples other than a few exceptions. In addition, the Cassie–Baxter model can only be applied for superhydrophobic surfaces under certain conditions. Vogler et al. suggested the cutoff for hydrophilic/hydrophobic should be θ ≈ 65° rather than 90° when considering long-range hydrophobic interactions [57]. Later, Jiang et al. proved the intrinsic wetting threshold for water should be 65°, which resulted from different hydrogen-bonded (hydrophilic)/non-bonded (hydrophobic) molecular water structures at the interface [58,59]. Thanks to the complicated situations of liquid motion on different kinds of surfaces, the debates will last for a long time and exploration of new models is still urgently needed.

## 3. Three-Dimensionally Printed Superhydrophobic Materials

Numerous strategies have been proposed in fabrication of superhydrophobic materials during recent decades, among which 3D printing techniques show unique advantages over others for replication of complex structures in a facile and accurate way. In the following section, we discuss the micro/nanofabrication of superhydrophobic objects through 3D printing. Different printing techniques, the pros and cons of them, and the specific methods are summarized.

### 3.1. Three-Dimensional Printing Technologies

Three-dimensional printing, also denoted as additive manufacturing (AM) technology, is a mature technology transforming 3D models into objects. The common steps for 3D printing are (1) design a 3D model through computer-aided design (CAD); (2) cut the model into slices with suitable thickness; (3) print the model layer by layer. The powerful technique has unparalleled advantages in the reproduction of objects with complex structures with high precision. Moreover, the printed products have wide applications ranging from the motor industry, dental science, and the construction sector to jewelry, aircraft, etc. [60]. Depending on different printing principles, 3D printing can be mainly divided into three categories, that is, photocuring printing techniques, including stereolithography (SLA) [61], digital light processing (DLP) [62], continuous liquid interface production (CLIP) [63], two-photon printing (TPP) [64] etc., jetting methods, such as inkjet printing (IJP) [65], direct ink writing (DIW) [66], and extruding strategies, such as fused deposition modeling (FDM) [67], and laser melting methods, such as selective laser sintering (SLS) [68], elective laser melting (SLM) [69], and many other newly developed techniques [70].

Although 3D printing has been successfully applied in fabricated superhydrophobic materials during the past few years, not all types of 3D printing can be used in this field due to the great challenge in construction of micro- and nanosized structures in high resolution with limited raw materials. The dominant 3D printing technologies that can be applied in biomimetic superhydrophobic materials are summarized in Figure 3 and Table 1. The advantages and disadvantages, typical raw materials, printing resolution, and printing speed are compared here. A brief introduction of commonly used 3D printing strategies is given as follows [71].

Stereolithography (SLA), also known as vat polymerization, is one of the earliest developed techniques in 3D printing [72,73,74,75]. Under ultraviolet (UV) light irradiation, light-induced polymerization occurs in a liquid photoresin container. The wavelength of UV light usually ranges from 365 to 405 nm. Following photopolymerization of the first layer, the remaining unreacted monomers continue to react with the irradiated liquid photoresins in the second layer. Thus, a solid 3D object can be fabricated through layer-by-layer curing of the resin. A post-curing process is commonly employed to promote conversion of unreactive monomers to improve the mechanical properties of printed objects. The resolution of SLA mainly depends on the size of the laser beam and, consequently, a high resolution of 100 nm can be realized in this system. SLA has great potential in printing objects with a large size and complex structures with sufficient precision.

The fabrication principle of digital light processing (DLP) is similar to SLA, but differently, a UV projector is used to project the image of the cross-section of an object into the liquid resin [62,76,77]. The core technology of DLP is determined by the optical semiconductor or digital microscope device (DMD) or DLP chip. Meanwhile, since the polymerization of the photoresins always takes place at the bottom of the vat, as a result, air is blocked and DLP is less sensitive to oxygen compared with SLA. More importantly, DLP can print objects with high precision yet limited size. Derived from common stereolithography, continuous microprinting strategies, such as continuous liquid interface production (CLIP), have been developed and applied for the fabrication of different objects [78,79,80]. In CLIP, an oxygen-permeable membrane is introduced to prevent radical polymerization. Similar to DLP, UV projection is located at the bottom, while a stable liquid area or “dead zone” can be maintained thanks to the oxygen, and continuous printing can be realized in this way. In consequence, the printing speed of CLIP is fast, up to 100 times faster than DLP. Furthermore, these microprinting strategies based on stereolithography have been successfully applied in construction of superhydrophobic materials during recent years.

Being one of the 3D printing techniques with the highest resolution (less than 100 nm), TPP employs a near-infrared femtosecond laser as a light source to initiate the photopolymerization process within a tight focal volume [81,82,83,84]. The focused laser spot can move freely within the resin, providing freedom in the fabrication of 3D objects. In addition, two-photon adsorption is linear only within the focused spot, and the intensity of two-photon adsorption is proportional to the square of the incident laser density. As a result, the precision of TPP can be very high. However, due to the deep permeation ability of the femtosecond laser source into transparent resin and difficulty of focusing inside metal supports, TPP is not suitable for such resin.

Inkjet 3D printing reproduces 3D objects on substrates via jetting and solidification of inkjet droplets [85,86,87,88]. The ink materials are ejected from head/nozzle either under certain pressure or through an actuator pulse generated from a thermal or piezoelectric head. The above process is repeated until completion of the desired objects. IJP is a powerful technique for the customized deposition of polymeric materials in an efficient and facile way, showing great potential in fabricating various kinds of structures. However, the printing resolution (50 μm) is low compared with other 3D printing methods due to the limitation of droplets, which may impede further application. The fabrication process of DIW is similar to IJP, where the ink droplets are extruded through the nozzle under external force and solidified along a specified path. The precision of this strategy mainly depends on the size of the nozzle, which is still small (fluctuating from 1–100 μm) compared with other 3D printing techniques.

In FDM, materials can be fed into and forced out through one or more heated nozzles/orifices, forming continuous filaments, following by solidification and adhesion to the preceding layer [89,90,91,92]. Typically, extruded materials are deposited onto a molding platform assisted by motors to move in 3D directions to create the desired shapes. Thermal plastic materials are used during FDM, which go through solid–liquid–solid phase transition. Numerous kinds of materials can be used in FDM, such as acrylonitrile butadiene styrene (ABS), polylactic acid (PLA), high-impact polystyrene (HIPS), thermoplastic polyurethane (TPU), etc. The printing resolution is dependent on the nozzle, reaching the highest resolution of about 100 μm. FDM is one of the most widely used prototyping methods in 3D printing due to a rich choice in designs and materials, low cost, and fast printing speed.

SLS/M is a kind of 3D printing method which is mainly applied to process solid powders. The main working mechanism is utilization of a high-power laser to sinter or melt powder and, after powder solidification, the desired 3D objects can be obtained by moving the molding platform. Throughout SLS, the processing temperature is kept high but less than the melting point of the powder [93,94,95]. Therefore, the particles are not completely melted, and the powder maintains its porosity. Particles can be reused during SLS, placed on the fabrication platform as support for the next step by a roller. In comparison, during SLM, particles are heated above the melting point, more particles are softened and melted to fill space, and the porosity is reduced [96,97,98]. The printing resolution of SLS/M can reach 50 μm, however, the particle size is commonly limited to 10 μm to prevent assembly and the majority of the materials employed in SLS/M show poor biocompatibility.

### 3.2. Three-Dimensional Printing of Biometric Superhydrophobic Materials

During recent years, large amounts of work have been reported on 3D printed biometric superhydrophobic materials thanks to their fast printing speed and high precision in the fabrication of complex structures. In general, these strategies can be categorized into the following groups: (i) 3D printing of special micro/nanostructures; (ii) post-modification of 3D printed objects; and (iii) 3D printing of bulk materials in one step, which will be illustrated in detail. For the micro/nanofabrication method, complex structures can be replicated directly to mimic the natural analogues, though these delicate objects may be fragile when it comes to friction or corrosion. While post-modification can be used as a powerful tool to further improve the anti-wetting properties of the printed objects, they may still suffer from destruction of the layers and loss of superhydrophobicity. In comparison, bulk materials can be fabricated directly and are resistant to possible wear and tear. However, they may be limited by the choice of suitable reaction systems and limited mechanical properties.

#### 3.2.1. Three-Dimensional Printing of Special Micro/Nanostructures

To start with, inspired by the lotus leaf, pillared micro/nanostructures have been printed by 3D printing [99]. Magdassi et al. fabricated superhydrophobic surfaces by DLP [100]. A novel ink was developed, mainly consisting of non-fluorinated acrylic monomers and dispersed hydrophobic fumed silica (HFS) (Figure 4a). The introduction of HFS brought about a certain surface roughness and lowered the surface energy. A cube with a micropillar array with pillar spacing of 300 μm and side length ranging from 70 to 130 μm on its faces could be fabricated. The best superhydrophobic property can be achieved with a pillar length of 70 μm and spacing of 300 μm, with CAs reaching 155° and sliding angle of 5°. In addition, thanks to the superhydrophobicity, the fabricated object could flow on the water surface freely, similar to that of legs of water striders. Credi et al. employed SLA to construct perfluoropolyether (PFPE) microstructures on flat substrates [101]. The surface tension of PFPEs was low enough. By further optimizing printing conditions, i.e., laser energy, cylindrical pillar arrays with diameter of 85 μm, height of 400 μm, and different spacings could be printed quickly (Figure 4b). For surfaces with pillar arrays (spacing of 200 μm) printed based on PFPE-tetracrylate oligomers, the CA was more than 150° and sliding angle ranged from 2°–5°.

Among all special 3D printed micro/nanostructures, re-entrant geometry has attracted extensive attention due to its efficiency in water repellency. In nature, springtails possess unique overhang structures, similar to mushrooms, which have been proven to be superhydrophobic [102,103,104]. The re-entrant structures were evaluated to be effective in supporting water droplets and keeping them away from the bottom by entrapping air [105] (Figure 5a). Several works have been reported based on 3D printed overhanging structures [29,30,31,106,107,108,109]. For instance, Sitti et al. fabricated mushroom-like double re-entrant poly (dimethylsiloxane) (PDMS) arrays based on a 3D printed template [106]. The arrays showed superb repellency to various kinds of liquid, even those with low surface tension. Gu’s group prepared both doubly and triply re-entrant structures through TPP, which were flexible enough to be printed on arbitrary substrates [31]. The triply re-entrant arrays have superrepellence to both water and organic liquids (Figure 5b). In another report of theirs, through further regulation of parameters such as diameter, distance, etc., three different adhesion states could be achieved [30]. Still, liquid repellence can be further improved by combining re-entrant mushroom structures with spring-like flexible arms [29]. Kinetic repellence to liquid can be realized by preventing liquid impalement, moreover, the contact time was reduced in this way (Figure 5c). To mimic cuticles of *Dicyrtomina ornate*, Mishra et al. innovatively built a wall of double re-entrant structures with one or two lateral caps on the sides, with centrally located normal double re-entrant arrays [109]. The addition of a secondary microstructure was proven to be effective for air entrapment and prevention of liquid movement laterally, showing superomniphobicity both in air and on submersion (Figure 5d).

Inspired by the ability of the *Salvinia molesta* leaf to form robust air cushions when immersed in water, several works have been reported on 3D fabrication of similar structures [32,33]. *Salvinia* possesses unique micro/nanostructures, microcellular arms can be found on top of the leaves, and groups of four arms are connected at the end, forming heterogeneous eggbeater structures [110,111,112]. The air retention ability, also known as the “*Salvinia* effect”, results from multiple factors thanks to the unique eggbeater structures, including structural support, maximized penetration energy, and pinning effect (Figure 6a). In 2015, Mattoli et al. fabricated *Salvinia*-like patterns by scaling down the original parameters from hydrophilic materials [32] (Figure 6b). However, a hydrophobic state can be achieved only with CAs around 120°. To further enhance the hydrophobicity, Chen’s group printed superhydrophobic eggbeater surfaces with a high-adhesion immersed surface through accumulation-based 3D (ISA-3D) printing [33] (Figure 6c). In ISA-3D, printed materials are accumulated along the light movement direction and multiscale objects can be printed in this way. In this work, to increase the surface roughness and mechanical strength of printed objects, multiwalled carbon nanotubes were added to liquid resins. The size of eggbeater arrays was similar to that of natural ones, showing superhydrophobicity (CAs~152°) and high adhesion, which can be regulated by changing the distance and number of connected arms.

Furthermore, in addition to the above structures, many other types of micro/nanostructures have been printed through various kinds of 3D printing techniques [113,114,115,116,117,118,119]. For instance, Wu et al. proposed a multiscale SLA technique, which can print objects with a large scale and high resolution [117]. The laser spot size can be adjusted flexibly with the assistance of a resonance grating filter. The layer thickness was also adjustable. As a proof of concept, samples with riblets and denticles were printed to mimic shark skin, and artificial lotus leaf can be fabricated accordingly (Figure 7a). Combining extrusion-based 3D printing with a robotic dispenser, PDMS arrays showing ratchet-like slip angle anisotropy were prepared [118] (Figure 7b). TCL could increase or decrease depending on whether the surface was tilted parallel to the slopes or not. Moreover, the larger TCL indicated stronger liquid–solid interactions, leading to an increase in slip angle to overcome the barrier. Tang et al. printed porous membranes with high water repellency and mechanical strength by DIW [119]. Nanosilica was added into the PDMS ink to improve mechanical strength (Figure 7c).

#### 3.2.2. Post-Modification of 3D Printed Objects

Under some circumstances, the 3D printed objects alone cannot realize superhydrophobicity due to the high surface energy and, in consequence, post-modification of these objects is of vital importance. One common modification method is coating hydrophobic materials onto the surfaces [34,35,36,120,121,122,123,124,125]. Motivated by hierarchical structures of rice leaf, Palza et al. firstly printed microchannels with intrinsic roughness by the SLA method, then the surface was modified by TiO_2_ nanoparticles [34] (Figure 8a). The wettability of these nanoparticles was functionalized by covalently connected hexadeciltrimethylsiloxane (HTMS) bonds. The original flat uncoated surface exhibited only hydrophobic behavior (CAs > 95°) with anisotropy of 35° and the average hysteresis was about 10°. For uncoated microchannels, the anisotropy increased. In comparison, for coated microchannels, a maximum of advancing contact angle of 165° with anisotropy of 5° and hysteresis less than 5° could be realized. A classic Cassie–Baxter state with air pockets trapped in these microchannel surfaces was verified both by direct experimental observation and numerical simulation results. Meanwhile, the coated surfaces exhibited self-cleaning property and UV irradiation-tuned wettability due to the existence of TiO_2_ nanoparticles.

In the Bruggen group’s work, the growth of two kinds of zeolitic imidazolate frameworks (ZIFs) with distinct morphology onto SLS printed polyamide (PA) porous membrane was reported [125] (Figure 8b). After coating PDMS, the composite surface presented a superhydrophobic state with static CAs reaching 158.6° and a sliding angle less than 2°. The morphology of the first ZIF layer was leaf-crossed structures, while the second layer of the ZIF was composed of flat rod-shaped and needle-like nanostructures. The unique hierarchical micro/nanostructures, combined with intrinsic rough PA membranes, contributed to the anti-wetting property of the objects, which can be further applied for highly efficient oil/water separation. Apart from the above hydrophobic coating, Yang et al. reported initiator-integrated 3D printing (i3DP) for surface post-modification [36]. In this method, a vinyl-terminated initiator is added into the photocurable resin, enabling further post-surface printing by the advantage of surface-initiated atom transfer radical polymerization (SI-ATRP) (Figure 8c). In this case, a cross-linked network was formed under UV irradiation, and the Br-containing initiator can induce ATRP for surface modification with any kind of polymers. Moreover, as a proof-of-concept, 1H,1H,2H,2H-perfluorodecyl methacrylate (PFMA) with low surface energy was chosen for further ATRP. Interestingly, after poly (PFMA) modification, the printed microlattices became superhydrohobic. In a similar way, microballs which can hold water perfectly without leakage can be printed and modified with poly (PFMA).

#### 3.2.3. Three-Dimensional Printing of Bulk Materials

In addition to the delicate design of complex structures and multistep post-modification strategies, during the past few years, several works focusing 3D printing of materials with bulk superhydrophobicity have been described [37,38,126,127]. Compared with other techniques, the main advantage of this method lies in the facile fabrication of bulk materials in fewer steps, while, when the surface structure is destroyed, the newly produced surface can show the same property. As a result, the printed objects are robust with increased resistance to external damage. A few typical cases will be illustrated in the following section.

Zhang et al. reported 3D printed microtextured polytetrafluoroethylene (PTFE) generated from a microprinting (μ-printing) method. First, PTFE nanoparticles were mixed with poly (ethylene glycol)diacrylate (PEGDA) in liquid resin and printed layer by layer [126]. This was followed by evaporation of water and sintering to decompose PEGDA, and pure and densely packed PTFE microstructures with nanostructured surfaces were obtained. The structures can be tuned by mixing ratio, water content, sintering temperature, etc. The optimized CAs can be as high as 167.3° with sliding angle less than 6°. Due to these properties, the printed structure can be applied in whispering-gallery-mode (WGM) resonators and electrostatic-driven biomimetic waterstriders. In the Helmer group’s work, superhydrophobic membranes with micro/nanoporosity were printed based on SLA in one step. Notably, the superhydrophobic property can be ascribed to several factors, employment of photocurable fluorinated resin, polymerization-induced phase separation (PIPS) resulting from the porogen mixture, variable size of nanopores by controlling the porogen ratio, etc. In addition, the objects were printed in a special “staircase” way, and a thin layer membrane showing superhydrophobicity (CAs~164°) could be easily peeled off. Mechanical stability tests of these superhydrophobic films showed that the membranes were robust and maintained water repellence even under multiple stretching or bending cycles.

Another successful example employing PIPS was reported by Levkin et al. [37]. In this case, superhydrophobic macroscopic objects with inherent nanoporosity were printed through DLP (Figure 9a). The liquid resin mainly consisted of hydrophobic (meth)acrylate monomers and porogen solvents and, during the photoinduced polymerization, bi-continuous 3D structures with two phases—more or fewer polymers—could be obtained after phase separation. The good/bad solvent ratio of porogen had a great influence on porosity, polymer globular size, as well as mechanical strength. Moreover, supercritical drying of the objects was a prerequisite to maintain nanoporosity. Objects with different complex structures can be prepared in this way (Figure 9b). Meanwhile, the scanning droplet adhesion microscopy (SDAM) method was used to measure the wetting property of these objects, through which the interactions between liquids and samples can be tested and recorded by a vertically mounted force sensor. A maximum advancing angle around 165° and receding angle of about 155° can be obtained after optimization. Additionally, thanks to the bulk nanostructures, the 3D objects were superhydrophobic even after 40 abrasion cycles. Both gas-permeable microfluidic devices and hierarchical porous oil-absorbent ones can be fabricated based on these 3D bulk materials, showing great potential for a variety of applications.

Recently, Su et al. fabricated large-scale superhydrophobic (half-a-meter sized) objects with abrasion resistance through an SLS strategy in one step [38]. Composite powders composed of hydrophobic fumed silica (HFS) and polymer (polypropylene (PP)) were printed. After sintering, a PP skeleton was formed, contributing to considerable roughness and mechanical strength (Figure 10a). With the further introduction of hydrophobic HFS, intrinsic superhydrophobicity was supposed to be achieved. In addition, the as-prepared thin film showed excellent abrasion-resistant stability under various kinds of wear damage tests. Moreover, beyond PP, the method can be applied for printing of superhydrophobic objects from diverse kinds of polymers, such as polyether block amide (PEBA), polyethylene (PE), polystyrene (PS), polymethyl methacrylate (PMMA), etc. Moreover, thanks to the ability of SLS in printing macrostructures (meter-sized), water-repellent aerial vehicles and abrasion-resistant shoes were printed accordingly (Figure 10b), proving the practical application potentials of this strategy in daily life.

## 4. Applications of 3D Printed Superhydrophobic Materials

On one hand, due to their excellent water-repellent properties, the 3D printed superhydrophobic objects can be applied in diverse fields, including photovoltaic [128,129,130], magnetic [131,132,133], and optoelectronic devices [128,129,130], detection platforms [134,135], cell culture [136,137], etc. [138]. However, due to the limited size, only a few typical application fields are introduced here, such as liquid manipulation [139,140,141,142], oil/water separation [123,127,143], drag reduction [140], anti-icing [144], etc. On the other hand, these objects show great potential in industrial applications thanks to the advantages of mature 3D printing techniques at a large scale and facile fabrication of complex objects with high precision.

### 4.1. Liquid Manipulation

A cactus can efficiently collect water from fogflow by using its conical spine clusters and belt-structured trichomes on the stem (Figure 11a), allowing it to survive in extremely dry environments [145,146]. Mimicking the cactus structures, Chen et al. prepared spine arrays with different branched shapes [147]. Benefiting from the ability of facile fabrication of complex structures of ISA-3D printing, natural branched clusters were successfully replicated (Figure 11b). In addition, the possible mechanism for highly efficient water collection was influenced by multiple factors. The hexagonally arranged clusters around 3D printed spines induced more turbulent moisture airflow, facilitating deposition of tiny water droplets. Meanwhile, the tilt angle and length of the spine can be easily optimized, generating Laplace pressure differences on it, which can further promote transportation of water droplets from the tip to the stem. Lastly, to further improve the collection of water, it was coated with a nanoscale superhydrophobic layer. In this case, the comprehensive effects of all the above factors contributed to efficient water collection. Shi and Zheng et al. fabricated cone arrays and the water-collecting performances were systematically studied [141]. The combination of a UV-induced controllable diffusion strategy and 3D printing was the key in successful fabrication of superhydrophobic cactus-like structures.

As an interesting model of inspiration, water can condense and transport along the peristome of *Nepenthes alata* in a unidirectional way with high speed [148,149,150]. Dong et al. mimicked this phenomenon by employing 3D printed flexible surfaces through an SLA and replication strategy [151]. Unidirectional rapid liquid transport can be realized in different paths under another immiscible liquid at a long range (Figure 11c). For an as-prepared 1D channel, liquid can spread directionally along the trajectory. Meanwhile, by bending and connecting straight lines, a 2D heart-shaped curved surface can be built. As can be seen, liquid can be transported in the curvilinear path unidirectionally. In order to investigate the influence of the curvature of the curved strip, 3D Möbius strip surface was fabricated by twisting and connecting the 1D strip. When placed under water, oil droplets can travel along both the 3D curved surface and surface of the 3D Möbius strip at an uneven rate in a single direction. In addition, liquid transported unidirectionally can be realized on a resin surface through a direct 3D printing method. The interesting unidirectional liquid transport on different kinds of surfaces will definitely inspire applications in fields such as microfluidics, soft robots, liquid electronics, etc.

**Figure 11 micromachines-14-01216-f011:**
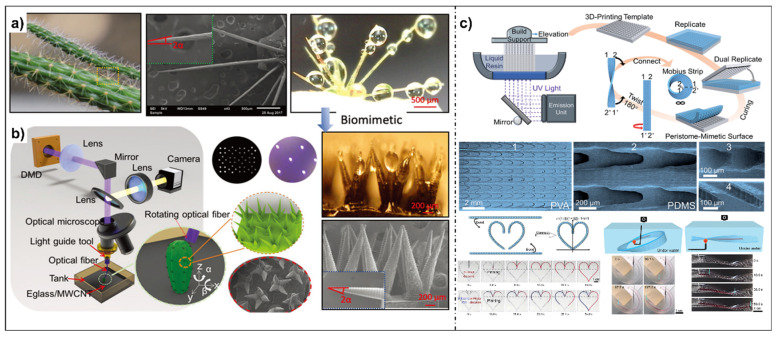
Three-dimensionally printed superhydrophobic objects applied in liquid manipulation. (**a**) Cactus covered with spines can be used for water collection. (**b**) Three-dimensionally printed artificial cactus-like arrays for highly efficient water collection based on ISA. Reproduced from [147], copyright 2019, WILEY-VCH Verlag GmbH & Co. KGaA, Weinheim. (**c**) Three-dimensionally printed surfaces mimicking *Nepenthes alata* with different morphologies. Unidirectional transport of liquid on these surfaces can have a linear trajectory in a one-dimensional channel, a two-dimensional curve along a heart-shaped pathway, and even occur along the three-dimensional infinite space. Reproduced from [151], copyright 2019, Royal Society of Chemistry.

### 4.2. Oil/Water Separation

Mei et al. reported fabrication of superhydrophobic/superoleophilic ceramic objects with gradient pore structures which can be effectively used for oil/water separation even under high temperatures (~200 °C) [152]. The objects can be prepared through a multistep process (Figure 12a). Firstly, a gradient pore ceramic structure (GPCS) was printed by DLP from ceramic slurry (Al_6_Si_2_O_13_). Then, the porous structure was sintered at about 1450 °C. After that, the GPCS was modified by SiO_2_ obtained from a sol–gel method and superhydrophobic/superoleophilic SiO_2_/GPCS with CAs reaching 162.1° was built in this way. The as-prepared SiO_2_/GPCS can be successfully applied in a series of oil/water separations, including kerosene, dichloromethane, hexadecane, soybean oil, diesel oil, etc. For all the test oils, SiO_2_/GPCS showed high separation efficiencies, all above 96.5%. Moreover, the separation efficiency and permeation flux can be further tuned by pore size and taper angles. In addition, the SiO_2_/GPCS showed excellent mechanical and chemical stability under ultrasonic treatment and in different solutions. Meanwhile, the oil/water separation ability of the SiO_2_/GPCS remained after several separation cycles, indicating high durability. Furthermore, as mentioned above, the SiO_2_/GPCS can efficiently separate oil from water in high-temperature environments.

A novel concept was proposed by Chan et al. using an all-3D printed superhydrophobic/oleophilic membrane for oil collection and recycling with confined surface tension and gravity [153]. The device can be divided into several parts, which can all be printed with different 3D printing techniques. For the central collection part, functional superhydrophobic graphene was printed onto porous nickel foam with laser-induced photochemical conversion of polyamide (Figure 12b). The composite membrane exhibited superhydrophobic (CAs~150°) and oleophilic (CAs~50°) properties. Then, the membrane was bent into a tube and a reservoir tube with suitable size was placed into it. A floating holder was printed through FDM from thermoplastic. The central part was integrated with the holder, and after assembly of six plastic foam pads, the self-floating oil recycling device was fabricated accordingly. The underlying mechanism of oil recycling of this device was that, due to the superhydrophobic/oleophilic property, only oil can permeate and drift into the recycling part by gravity. As time goes on, all surrounding oil can be collected. The above proposal was proved by relevant experiments. The novel concept for oil collection has advantages of high reusability in an environmentally friendly way. Still, it is challenging for such a device to be used in real wastewater as the conditions are much harsher.

### 4.3. Drag Reduction

In nature, the surface of shark skin is covered with numerous dermal denticles, and the shape and size of these denticles vary with location [154]. This unique structure is found to be effective in reducing formation of vortices, leading to facile water movement around the skin (Figure 13a). Inspired by this, extensive works have been carried out on fabrication of objects with drag reduction ability (Figure 13b). Meanwhile, superhydrophobic surfaces with certain roughness and low adhesion have been proven to be useful in drag reduction [155,156]. Still, there is controversy on the role of superhydrophobic surfaces regarding whether they reduce or increase drag. Two typical models are proposed for possible drag reduction of superhydrophobic structures. One is the concept of a slip boundary condition proposed by Navier [157]. According to this model, the vortex cushion effect can be generated when water flows past hydrophobic smooth surfaces, leading to wall slip. The wall slip will lead to a decrease in interfacial velocity and the sheer force, which contributes to the drag reduction. Another model is the plastron effect proposed by McHale et al. [156], in which air layer can be formed and stabilized when water contacts superhydrophobic surfaces, which can effectively reduce the drag.

Zhang et al. fabricated superhydrophobic (CAs~160°) petal-like objects with microstructures printed by the projection μ-SLA method [140] (Figure 13c). Moreover, the arch-shaped arrays mimicked the Nepenthes peristome, possessing similar excellent liquid pinning ability, and the tested water droplet could stay firmly on top even when the surface was upside down. The underlying mechanism of this phenomenon can be ascribed to a combination of the arch-shaped curve effect and sharp-edged effect of the printed structures, which play vital role in generating energy barriers and constraint forces to restrict the spread of water droplets. Moreover, the petal-like structures surrounded by walls can be applied in bubble capture and drag reduction. When placed under water, stable bubbles can be captured inside the microstructures (Figure 13d). According to the results, drag force can be reduced greatly with the as-prepared objects, especially for flow with high velocity (Figure 13e). The main reason for drag reduction in this case was related to the captured bubbles, which can separate water from the solid surface, leading to a boundary slip effect to reduce fluid drag.

### 4.4. Potential Application in Anti-Icing

Ice accretion on surfaces can be hazardous to aircrafts, power and communications systems, and energy transport, causing risks to human life and production. In consequence, numerous kinds of strategies have been proposed over recent years for the development of anti-icing and de-icing systems [144]. Among these, biomimetic superhydrophobic surfaces (SHSs) inspired by the “lotus leaf effect” and slippery liquid-infused porous surfaces (SLIPSs) inspired by Nepenthes have shown great potential in anti-icing applications [158,159] (Figure 14a). The mechanism of anti-icing SHSs and SLIPSs can be summarized as follows: (1) before icing, water can be removed directly due to the superhydrophobicity of the surfaces. Moreover, the condensed water droplets can jump from the surface in a self-propelled way, resulting from both micro/nanostructures and the self-cleaning ability of the surfaces [160]. In addition, water droplets can slip easily on SLIPSs compared with SHSs due to reduced interfacial tension. (2) During the icing process, these surfaces were proven to be effective in prolonging the icing delay time (DT) and delaying crystallization nucleation effectively [161]. For SHSs, the captured air can provide thermal resistance to decrease heat transfer, reducing the contact area and preventing nucleation. For SLIPSs, the ice point can be reduced due to the polymers released from the surface, inhibiting ice nucleation and phase transition.

For instance, Pan et al. built tri-scale micro/nano-SHSs with excellent anti-icing performances and icephobic properties [162] (Figure 14b). In this case, ultrafast laser ablation assisted with chemical oxidation was utilized. The hierarchical structures were composed of microcone arrays covered with grass-like nanostructures and flower-like microstructures. The surface showed superhydrophobic property (CAs~161°) with low SA (~0.5°) and critical Laplace pressure as high as 1450 Pa. Impacting droplets can rapidly roll off the SHS. In addition, due to the multiple effects of self-propelled jumping of the condensed droplets, hierarchical condensation, the SHS can be applied in high humidity. Finally, thanks to the presence of stable air pockets, the heterogeneous nucleation at the solid–liquid interface was delayed. Although, to date, little work has been reported on using superhydrophobic 3D printed objects for anti-icing, the as-prepared objects have shown great potential in this field. Most importantly, the 3D printing technique can produce SHSs and SLIPSs with tunable properties in a facile way. As mentioned above, different kinds of SHS and SLIPS can be fabricated through diverse 3D printing methods with high precision. As a result, there are good reasons to believe that, in the near future, robust ice-resistant superhydrophobic and slippery liquid-infused porous surfaces will be successfully fabricated from 3D printing.

## 5. Conclusions and Outlook

### 5.1. Current Progress

In this review, we briefly introduced current works based on 3D printed superhydrophobic materials. The wetting regimes, inspired by two famous phenomena, the “lotus leaf effect” and “rose petal effect”, were introduced by describing several basic models. Through relevant description, the definition of superhydrophobicity and the determining factors were discussed. In the followed section, the basic mechanism of 3D printing and commonly employed techniques were illustrated. The principle, advantages, and limitations of various printing methods, including SLA, DLP, TPP, IJP, DIW, FDM, and SLS/M, were introduced. The above-mentioned techniques can be successfully applied in fabrication of superhydrophobic objects in diverse ways, ranging from printing of unique micro/nanostructures and post-modification to printing of bulk materials. Several typical structures, including pillar structures, re-entrant structures, and eggbeater structures can be fabricated by different printing techniques. In addition, by modifying the printed objects with superhydrophobic coating, anti-wetting can be achieved with different materials. In addition, printing bulk materials with inherent superhydrophobicity have attracted extensive attention due to their stability and resistance. Thanks to the excellent water repellency, the as-prepared objects can be used for liquid manipulation, oil/water separation, as well as drag reduction. Furthermore, thanks to the merits of facile operation of 3D printing in construction of SHSs and SLIPSs, the printed surfaces show great potential in anti-icing applications, which have not been reported yet.

### 5.2. Challenges and Perspectives

Despite all the great progress made in this area, urgent challenges still remain.

(1)For one thing, the “structure–performance” relationship is absent in the current field. As a result, to solve this problem, computer-assisted design of structures with superhydrophobicity should be taken into consideration. Various computational methods, such as density function theory (DFT), molecular dynamics (MD), ab initio molecular dynamics (AIMD), and finite element analysis, should be used together for construction of “structure–performance” relationships. With the aid of these methods, machine learning, which can improve screening efficiency for suitable structures, should be applied in future 3D printing of superhydrophobic objects.(2)In addition, printing of bulk materials in one step may be one of the main research hot spots due to the inherent superhydrophobicity that is not limited to the surfaces. To realize facile fabrication of such structures, currently developed photopolymerization-induced microphase separation (PIMS) systems based on reversible addition–fragmentation chain transfer (RAFT) polymerization with tunable micro/nanoscale structures with improved mechanical stability show great potential in 3D printing of superhydrophobic objects. In PIMS, the chain of a macromolecular chain transfer agent (macro-CTA) is extended via RAFT polymerization, generating block copolymers with thermodynamically incompatible block segments. The PIMS method has advantages for the facile fabrication of objects with rapid speed, excellent robustness, and precise control of morphologies, which can be applied in future fabrication of superhydrophobic materials through photo-induced 3D printing strategies, i.e., DLP and SLA.(3)Moreover, in addition to traditional 3D printing, four-dimensional (4D) printing of “smart” materials has attracted attention since 2013. Dynamic structures with tunable shape, property, or functionality responding to stimuli can be printed by 4D printing. The external stimuli can be light, temperature, pH, etc. Shape-memory materials, which can be applied directly, multimaterial integration, and mathematical modeling-guided design of deformation energy based on stress mismatch between two layers are the three main strategies in 4D printing. The application of 4D printing in fabrication of superhydrophobic materials will definitely provide new ideas in generating smart devices with multifunctional structures to meet increasing demands in wider applications.(4)In addition, due to the capability of 3D printing in facile customizable design of meter-sized objects through, e.g., SLS/M, the printed superhydrophobic materials could be used in practical applications in industry and daily life in the near future.(5)Last, but most important, more bioinspired surfaces beyond superhydrophobicity can be built from 3D printing, leading to further understanding of natural phenomena and construction of “structure–performance” relationships depending on different surfaces, inspiring the design of brand-new devices for wider applications.

**Scheme 1 micromachines-14-01216-sch001:**
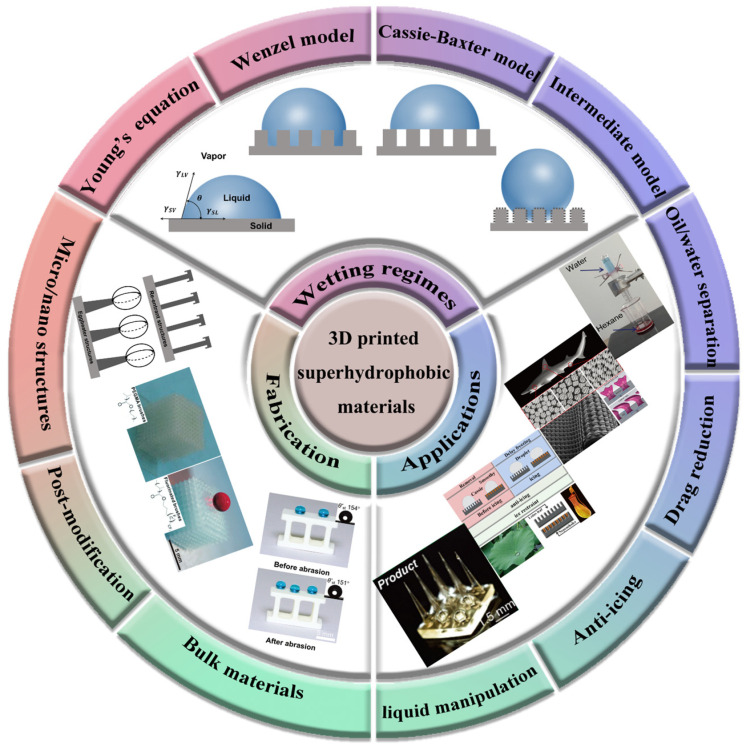
Schematic illustration of 3D printed superhydrophobic materials, including different wetting regimes, general fabrication strategies, reproduced from [36], copyright 2013, Royal Society of Chemistry, reproduced from [37], copyright 2021, WILEY-VCH Verlag GmbH & Co. KGaA, Weinheim and applications, reproduced from [139], copyright 2021, American Chemical Society, reproduced from [154], copyright 2014, the Company of Biologists, reproduced from [152], copyright 2021, Elsevier.

## Data Availability

Not applicable.

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
