# Peer review of "Biomimetic Superhydrophobic Materials through 3D Printing: Progress and Challenges"

_micromachines, 2023, doi:10.3390/mi14061216_

Round 1

Reviewer 1 Report

The paper introduces the progress and challenge of Biomimetic superhydrophobic materials through 3D printing, in detail elaborates the natural superhydrophobic surface and wetting mechanism, 3D printing technology successfully applied in manufacture of superhydrophobic case of the object, and the challenge of are discussed. Overall, it is a topic of interest to the researchers in the related areas but the paper needs very significant improvement before acceptance for publication. My detailed comments are as follows:

1.       Whether there is a correlation between wetting regimes, different manufacturing techniques, and various applications of printed objects. What is the meaning of exploring wetting mechanisms for the application of superhydrophobic properties or 3D printing technology?

2.       What are the consequences of “the ‘structure-performance’ relationship is absent in current field.” and Are there any studies on relevant structural properties? If so, what efforts have been made to achieve the effects and limitations?

3.       The accurate definition of the concept of superhydrophobic has not been found.

4.       The transition between chapters should be improved

5.       Please carefully check the spelling and writing of the article, for example, confirm that "3. D "is written correctly in section 3, line 210.

6.       The applications of superhydrophobic materials can be listed more comprehensively.

Please carefully check the spelling and writing of the article.

Author Response

Response to Reviewers' Comments and Revised Details

-------------------------------------------------------------------------------------------------------

Reviewer 1

Comment 1: Whether there is a correlation between wetting regimes, different manufacturing techniques, and various applications of printed objects. What is the meaning of exploring wetting mechanisms for the application of superhydrophobic properties or 3D printing technology?

Response 1: Thank the reviewer for the professional comments. We believe that studying of wetting regimes is of vital importance as guidelines for fabrication of superhydrophobic objects. For instance, the wetting mechanism for re-entrant structures is special (Figure. 5a), combining unique structures and support, penetration energy and pinning effect, which can be used in construction of superhydrophobic materials through 3D printing (Figure. 5b-d). Furthermore, these objects can be applied in anti-wetting field under different situations, i.e., underwater (Figure. 5d) or high repellence to organic liquids (Figure. 5b) due to their unique wetting mechanisms.

Comment 2: What are the consequences of “the ‘structure-performance’ relationship is absent in current field.” and Are there any studies on relevant structural properties? If so, what efforts have been made to achieve the effects and limitations?

Response 2: Thank the reviewer for this valuable question. The “structure-performance” relationship is absent in superhydrophobic materials and few studies can be found in this field. In fact, up to now, we haven’t found any relevant systematic studies in this field. Besides, we believe such studies are very important in fabrication of superhydrophobic objects through 3D printing. On one hand, the clear “structure-performance” relationship can provide direct guidance to fabrication of superhydrophobic materials. To be specifically, during the construction process, the raw materials, special micro/nano structures, and fabrication techniques can be chosen more rationally according to the “structure-performance” relationship. Through which the time and cost can be reduced accordingly. Furthermore, such “structure-performance” relationship is of vital importance to meet the increasing demands for customized products for different applications. To sum up, more efforts should be made to construct “structure-performance” relationship in this field.

Comment 3: The accurate definition of the concept of superhydrophobic has not been found.

Response 3: Thank the reviewer for the kind suggestions. We have added the accurate definition of superhydrophobicity in “Introduction” part in the revised manuscript and highlight it.

Comment 4: The transition between chapters should be improved.

Response 4: Thanks for the reviewer’s valuable suggestion. According to your suggestion, we have modified the transition between chapters in the revised manuscript to make it smooth and readable.

Comment 5: Please carefully check the spelling and writing of the article, for example, confirm that "3. D "is written correctly in section 3, line 210.

Response 5: Thanks for the reviewer’s the kind suggestions. There may be some mistakes after the manuscripts is uploaded and edited, which is different from the original one. We have carefully checked and amended all the spelling in the revised manuscript to avoid similar mistakes.

Comment 6: The applications of superhydrophobic materials can be listed more comprehensively.

Response 6: We appreciate these suggestions. We have modified the relevant part in the revised manuscript more comprehensively, and several references are added. However, the superhydrophobic materials can be applied in various kinds of fields, which cannot be all listed in our manuscript due to the limited size. Consequently, we give a summary of all possible applications of superhydrophobic objects and put emphasis on several typical application fields in the manuscript.

Reviewer 2 Report

 This review provides a comprehensive overview of how to fabricate artificial superhydrophobic materials through 3D printing. The review covers different wetting models and various 3D printing techniques used to fabricate superhydrophobic structures. It also highlights the applications of these materials, such as liquid manipulation and oil/water separation. Finally, the authors suggest using computational methods and advanced printing techniques, like photo-polymerization-induced microphase separation and four-dimensional printing, to overcome challenges in this field. This is an informative review, and we recommend a minor revision before its final acceptance. Several issues that need to be considered are listed as follows.

 1.       The title " Biomimetic superhydrophobic materials " leaves a deep impact on the readers. Hence, what is worth studying about the structure and function of superhydrophobic materials in constructing artificial analogs?

 2. In the review, the authors should give a comparison of different 3D strategies for the fabrication of superhydrophobic objects. For example, what are the main advantages or disadvantages of them?

 3. Several other valuable works need to be cited to help readers better understand biomimetic superhydrophobic materials fabricated through 3D printing. For example, potential applications of 3D printed superhydrophobic objects in magnetic devices have been systematically studied (ACS Appl. Mater. Interfaces 2023, 15, 23971–23979; Addit. Manuf., 2023, 69, 103542; Chem. Eng. J., 2023, 463, 142388; Adv. Mater. 2022, 34, 2203814.).

Author Response

Response to Reviewers' Comments and Revised Details

-------------------------------------------------------------------------------------------------------

Reviewer 2

Comment 1: The title "Biomimetic superhydrophobic materials " leaves a deep impact on the readers. Hence, what is worth studying about the structure and function of superhydrophobic materials in constructing artificial analogs?

Response 1: Thank the reviewer for the professional comments. In fact, learning from nature is one of the most instructive objects, ranging from the mechanism of natural phenomenon to fabrication and application of artificial analogues. Superhydrophobic materials possess excellent water-proof properties, and unique micro/nano structures have inspired extensive research in this field. However, currently, in the study of constructing artificial superhydrophobic objects, the “structure and function” relationship which can be used as guideline is still absent. Consequently, it’s worth to fabricate the relevant “structure and function” relationship libraries to know exactly which kinds of materials and structures can be used in superhydrophobic materials, and how to tune these structures to meet the increasing demands in various application fields.

Comment 2: In the review, the authors should give a comparison of different 3D strategies for the fabrication of superhydrophobic objects. For example, what are the main advantages or disadvantages of them?

Response 2: Thank the reviewer for the valuable question. We have added a summary of the main advantages or disadvantages of comparison of different 3D strategies for the fabrication of superhydrophobic objects in section “3.2 3D printing of biometric superhydrophobic materials” in the revised manuscript according the reviewer’s suggestion.

Comment 3: Several other valuable works need to be cited to help readers better understand biomimetic superhydrophobic materials fabricated through 3D printing. For example, potential applications of 3D printed superhydrophobic objects in magnetic devices have been systematically studied (ACS Appl. Mater. Interfaces 2023, 15, 23971–23979; Addit. Manuf., 2023, 69, 103542; Chem. Eng. J., 2023, 463, 142388; Adv. Mater. 2022, 34, 2203814.).

Response 3: Thanks for the reviewer’s the kind suggestions. We have accommodated his/her comment by revising the manuscript. We have added the above references in the revised manuscript. According to the reviewer’s comment in the application part, references [131-133], [138] were added.

Reviewer 3 Report

The current manuscript presents a review related to biomimetic superhydrophobic materials through 3d printing. The manuscript is well-written and organized. However, some minor issues should be considered as follows:

- The numbering of the subsections should be carefully revised. 

- The term "3d printing" is missing the number 3 in most subsections' titles.

- The conclusion section should be focused, a pullet points style is recommended to be used to summarize the progress in the current title, current challenges, ad the future perspective. 

.

Author Response

Response to Reviewers' Comments and Revised Details

-------------------------------------------------------------------------------------------------------

Reviewer 3

Comment 1: The numbering of the subsections should be carefully revised.

Response 1: Thanks for the reviewer’s the kind suggestions. We have carefully checked the numbering of the subsections according to your suggestion.

Comment 2: The term "3d printing" is missing the number 3 in most subsections' titles.

Response 2: Thank the reviewer for the valuable suggestions. This may be caused by the format conversion problems after upload of the manuscript. We have modified all these mistakes in the revised manuscript.

Comment 3: The conclusion section should be focused, a pullet points style is recommended to be used to summarize the progress in the current title, current challenges, and the future perspective

Response 3: Thanks for the reviewer’s the kind suggestions. We have rearranged the conclusion section in a more organizable way according to your suggestion, current progress, challenges and perspectives are summarized in sequence in the revised manuscript.
